# Yeasts as Complementary Model Systems for the Study of the Pathological Repercussions of Enhanced Synphilin-1 Glycation and Oxidation

**DOI:** 10.3390/ijms22041677

**Published:** 2021-02-07

**Authors:** David Seynnaeve, Daniel P. Mulvihill, Joris Winderickx, Vanessa Franssens

**Affiliations:** 1Functional Biology, Division of Molecular Physiology of Plants & Microorganisms, KU Leuven, 3000 Leuven, Belgium; david.seynnaeve@kuleuven.be (D.S.); joris.winderickx@kuleuven.be (J.W.); 2School of Biosciences, Ingram Building, Room 424, University of Kent, Canterbury, Kent CT2 7NH, UK; d.p.mulvihill@kent.ac.uk

**Keywords:** yeast, Parkinson’s disease, neurodegeneration, protein aggregation, Synphilin-1, glycation, oxidation

## Abstract

Synphilin-1 has previously been identified as an interaction partner of α-Synuclein (αSyn), a primary constituent of neurodegenerative disease-linked Lewy bodies. In this study, the repercussions of a disrupted glyoxalase system and aldose reductase function on Synphilin-1 inclusion formation characteristics and cell growth were investigated. To this end, either fluorescent dsRed-tagged or non-tagged human *SNCAIP*, which encodes the Synphilin-1 protein, was expressed in *Saccharomyces cerevisiae* and *Schizosaccharomyces pombe* yeast strains devoid of enzymes Glo1, Glo2, and Gre3. Presented data shows that lack of Glo2 and Gre3 activity in *S. cerevisiae* increases the formation of large Synphilin-1 inclusions. This correlates with enhanced oxidative stress levels and an inhibitory effect on exponential growth, which is most likely caused by deregulation of autophagic degradation capacity, due to excessive Synphilin-1 aggresome build-up. These findings illustrate the detrimental impact of increased oxidation and glycation on Synphilin-1 inclusion formation. Similarly, polar-localised inclusions were observed in wild-type *S. pombe* cells and strains deleted for either *glo1^+^* or *glo2^+^*. Contrary to *S. cerevisiae*, however, no growth defects were observed upon expression of *SNCAIP*. Altogether, our findings show the relevance of yeasts, especially *S. cerevisiae*, as complementary models to unravel mechanisms contributing to Synphilin-1 pathology in the context of neurodegenerative diseases.

## 1. Introduction

Lewy body inclusions and Lewy neurites are important hallmarks of neurodegenerative diseases such as Parkinson’s disease (PD) and Lewy body dementia. They consist primarily of α-Synuclein (αSyn) aggregates and occur in different neuronal cell populations including dopaminergic neurons [1]. In addition to αSyn, many other proteins have been detected in Lewy bodies, including the αSyn-interacting protein Synphilin-1 (SY1) [2,3,4,5]. SY1 is a 919 amino acid long protein encoded by the αSyn-interacting protein (*SNCAIP*) gene, which is expressed in a variety of tissues and enriched in the brain.

SY1 is predominantly expressed in neurons and present at presynaptic nerve terminals [6], and it has been proposed to be involved in regulating synaptic function together with αSyn, possibly by being anchored to the vesicle membrane via an interaction with αSyn [6]. The pathological repercussions of heterologous expression of human *SNCAIP* were studied in a variety of model organisms, including mice models [7,8,9,10,11], Drosophila [12], neuroblastoma cells [13], and yeast [14]. Expression of human *SNCAIP* in wild-type (wt) *S. cerevisiae* strain resulted in the formation of SY1-containing inclusions, starting with the formation of foci at endomembranes and lipid droplets. Büttner et al. speculated that the interaction with lipid rafts could facilitate dimerisation and self-assembly of SY1 [14]. In exponentially growing cells, SY1 did not induce any growth defects. This was attributed to polarisome-mediated retrograde transport [15] of said inclusions from daughter to mother cells, in which protective aggresomes can be formed and degraded. In aged cells, on the other hand, *SNCAIP* expression reduced survival and triggered apoptosis and necrotic cell death in a Sir2 dependent manner, most likely due to excessive aggresome build-up during growth and consequential deregulation of autophagic processes. In a co-expression model, SY1 aggravated αSyn aggregation, which was correlated with enhanced αSyn phosphorylation on Ser_129_ [14].

Glycation has primarily been studied in the context of diabetes mellitus, in which dysregulation of glucose metabolism results in the eventual formation of advanced glycation end-products (AGEs) via the non-enzymatic Maillard reaction [16,17]. These AGEs are primarily generated as a result of glucose metabolism by-products reacting with the free amino-groups of arginine and lysine residues of proteins. AGEs and AGE-modified proteins (cross-linked) exert toxicity via binding to a variety of receptors, including the receptor for AGE, termed ‘RAGE’, thereby increasing oxidative stress and inflammation through the formation of reactive oxygen species (ROS) and the induction of NF-κB [18,19,20,21,22]. ROS can then further facilitate the formation of AGE in a positive feedback loop [23]. Methylglyoxal (MGO), a by-product of glycolysis, is the most reactive glycation agent and is catabolised primarily via the glyoxalase system, comprised of Glo1 and Glo2, and by aldose reductases (e.g. Gre3) (Figure 1 and Table 1) [24,25]. The glyoxalase system is ubiquitous in all organisms so far examined, with the glyoxalase system in baker’s yeast first described by Racker [26]. The glyoxalase I (*glo1*^+^) from the fission yeast *Schizosaccharomyces pombe* shares 50% amino acid sequence identity with the *Saccharomyces cerevisiae* Glo1 [27]. *S. cerevisiae* has two isoforms of the structural genes for glyoxalase II, *GLO2* and *GLO4*. Expression of *GLO2* occurs both in glucose and glycerol media, while *GLO4* is expressed only in glycerol medium [28]. The identified *S. pombe glo2*^+^ gene has not yet been experimentally validated [29]. Contrary to *S. cerevisiae*, *S. pombe* does not possess a Gre3 ortholog. Aldose reductases such as Yak3/YakC have been identified in *S. pombe*, but none of these have reported MGO-detoxifying activity [30,31].

The observation of AGEs in Lewy bodies in cases of incidental Lewy body disease suggested that AGEs might trigger Lewy body formation in individuals that are considered pre-PD patients [32]. In addition to these observations made in humans, AGEs were also detected in dopaminergic neurons of a 1-methyl-4-phenyl-1,2,3,6-tetrahydropyridine (MPTP) mouse model of parkinsonism. Despite these observations, as yet there is no direct evidence that glycation is causally related to PD. Nonetheless, the influence of glycation on the aggregation of αSyn has been documented in a variety of organisms [33,34,35].

In this research article, we add to the findings described above by assessing the repercussions of a disrupted glyoxalase system and aldose reductase Gre3 activity on SY1 inclusion formation and SY1-mediated growth effects. To this end, we employed a genetic approach by analysing *S. cerevisiae* deficient in the *GLO1*, *GLO2*, and *GRE3* genes and *S. pombe* strain deficient in *glo1*^+^ and *glo2*^+^.

## 2. Results

### 2.1. SY1 Production Results in a Growth Defect of S. cerevisiae Strains Devoid of Glo2 and Gre3

First, the impact of a disrupted methylglyoxalase system and aldose reductase activity on the SY1-induced growth phenotype was examined. To this end, wt, *glo1*∆, *glo2*∆, and *gre3*∆ *S. cerevisiae* strains transformed with either an empty vector (EV) or a plasmid expressing native *SNCAIP* were subjected to growth analysis. These results revealed that both wt and *glo1*∆ cells containing the EV grow equally well (Figure 2A), despite the level of MGO in log-phase *glo1*Δ cells being approximately two times higher than that in wt cells [29]. Furthermore, *glo2*∆ and *gre3*∆ strains expressing the EV construct grew slower than wt and *glo1*∆ strains (Figure 2A). While wt and *glo1*∆ strains enter stationary growth phase around optical density (OD) ±0.8, *glo2*∆ and *gre3*∆ strains only reached an OD_max_ of ±0.6 (Figure 2A). The lack of repercussions on the growth of disrupting Glo1 functionality indicates that lack of glutathione-dependent methylglyoxalase activity can be compensated for by Gre3 activity, at least in conditions in which the cells are only subjected to endogenous MGO levels. On the contrary, disrupting Glo2 activity strongly affected cell growth. This finding contrasted with earlier findings showing no difference in growth between wt, *glo1*∆, and *glo2*∆ strains in a minimal medium containing 2% glucose [28]. This could be attributed to different growth conditions used (4 mL liquid medium in tubes versus 200 µL in a 96-well plate in this study), which can affect the oxygen availability and respiration efficiency. Finally, the strongly affected growth ability of *gre3*∆ cells in comparison to wt and *glo1*∆ cells indicates that Gre3-mediated conversion of endogenous MGO to acetol is necessary to prevent cellular damage caused by endogenous MGO-induced oxidative stress and AGE formation. Again, this contrasted with previous findings [31] indicating no growth difference between wt and *gre3*∆ cells, as assessed via spot test analysis on a minimal medium containing 2% glucose.

The genetic background in which the *GRE3* deletion was introduced (BY4741 in this study vs W303 in [31]) could explain the discrepancy in growth ability observed. In line with results obtained by [14], *SNCAIP* expression in wt and *glo1*∆ strains did not induce a growth defect during the exponential growth phase. In contrast, the growth of the *SNCAIP*-expressing *glo2*∆ and *gre3*∆ strains was clearly reduced compared to the EV control. These findings were confirmed by calculating the ∆T_50_, which indicates the number of hours needed for a strain to reach the OD value corresponding to half the OD_max_ value, and by performing a spot test analysis (Figure 2A right and Figure 2C left). The observed toxicity upon expressing *SNCAIP* in the *gre3*∆ strain indicates growth repercussions associated with enhanced glycation of SY1. Indeed, SY1 is highly susceptible to glycation due to the presence of 38 arginine and 71 lysine residues.

While Glo1 converts the spontaneously formed MGO-glutathione complex into S-D-lactoylglutathione, Glo2 is responsible for the conversion of S-D-lactoylglutathione into glutathione and D-lactate. As such, Glo2 recycles the cytoplasmic pool of glutathione. Cytoplasmically generated glutathione can either remain in the cytosol or be relocated into the mitochondria via active transport. The mitochondrial pool of glutathione is considered the major redox system in numerous processes, such as the preservation of sulfhydryl groups of mitochondrial proteins in the appropriate redox state and protecting mitochondrial DNA and membrane integrity against ROS-induced oxidative stress. Disruption of cytoplasmic Glo2 activity results in S-D-lactoylglutathione accumulation and consecutively keep glutathione trapped in the cytosol. This can then lead to decreased glutathione availability in the mitochondria, disrupting the glutathione-dependent antioxidant enzyme system [36,37].

Therefore, we reason that the observed toxicity in *SNCAIP*-expressing *glo2*∆ cells is associated with increased SY1 oxidation. The impact of *SNCAIP* expression on growth was also assessed in wt, *glo1*∆, and *glo2*∆ fission yeast strains. *S. pombe* does not possess a *GRE3* homolog and glyoxalase 2 activity of the proposed *S. pombe GLO2* homolog, *glo2*^+^, has yet to be established [29]. Contrary to our findings in *S. cerevisiae*, no growth differences between EV-expressing strains, nor a *SNCAIP* expression-dependent inhibitory effect on growth was observed in any strain examined (Figure 2B). These results were consistent with ∆T_50_ values and spot test analyses (Figure 2B right and Figure 2C right).

### 2.2. Disrupting Methylglyoxalase System and Aldose Reductase Activity Increases the Number of Large dsRed-SY1 Inclusions in S. cerevisiae

In wt *S. cerevisiae*, SY1 is initially cytoplasmic, which subsequently forms small oligomers and larger membrane-associated aggregates (Figure 3A). Despite this, *SNCAIP* expression does not impact growth kinetics in wt *S. cerevisiae* [14] (Figure 2A). Because *SNCAIP*-expressing *glo2*∆ and *gre3*∆ cells were characterised by reduced growth, we hypothesised that these strains could display altered inclusion formation. Indeed, a functionally disrupted glyoxalase system could increase glycation, which has been shown to compete with ubiquitination and consequent proteasomal degradation [35] of SY1, thereby promoting its accumulation and inclusion formation. Therefore, SY1 inclusion size and number were analysed in the *glo1*∆, *glo2*∆, and *gre3*∆ *S. cerevisiae* strains. Although fewer *glo2*∆ and *gre3*∆ cells displayed fluorescence (Figure 3B), a higher percentage of *glo2*∆ and *gre3*∆ cells contained SY1 inclusions upon comparison with wt and *glo1*∆ cells (Figure 3C). The percentage of wt cells displaying inclusions was in line with previous observations reported by [14]. Similar to wt, SY1 mainly formed large aggregates in *glo1*∆, *glo2*∆, and *gre3*∆ strains (Figure 3D). A striking difference between these strains, however, was the number of cells with numerous large aggregates. Significantly more *glo2*∆ and *gre3*∆ cells contained two or more large aggregates, rather than only one large inclusion. Contrary, an equal number of wt and *glo1*∆ cells displayed either one or two or more large inclusions (Figure 3E). Thus, increased SY1-mediated toxicity in *glo2*∆ and *gre3*∆ cells is associated with an increased number of cells displaying multiple large aggregates. Indeed, both the formation of disulphide bridges as a consequence of enhanced oxidation of cysteine residues, and the formation of cross-links between lysine and arginine residues through increased glycation could promote conformational changes and subsequent inclusion formation [33,34,35,38].

### 2.3. Immunofluorescent Detection of SY1 in S. pombe Reveals Polar-Localised Inclusions

Similarly, we assessed SY1 inclusion formation in wt, *glo1*∆, and *glo2*∆ *S. pombe* strains. To this end, we expressed non-tagged *SNCAIP* and detected using an anti-SY1 antibody. SY1 inclusions were observed in cells of all three strains, with inclusion-positive cells containing either one polar-localised inclusion or two inclusions, one at each of the two poles (Figure 4A). In symmetrically dividing organisms such as *S. pombe*, asymmetric distribution of fused protein aggregates between daughter cells has been shown to serve as a protective mechanism that allocates damaged proteins to only one daughter cell, while preserving the cellular health of the other daughter cell [39,40,41]. Quantification did not reveal a statistically significant difference in the number of SY1 inclusion-positive cells nor in the number of cells displaying solely one unipolar or two bipolar inclusions between any of the strains (Figure 4B). However, the inclusion size in cells containing two inclusions was highly similar to the size of inclusion in cells that only contained one inclusion.

Therefore, it seemed unlikely that the observation of one inclusion is the result of the stress-induced fusion of multiple smaller inclusions and that its asymmetric distribution serves a protective purpose. We concluded that the induced stress associated with the inheritance of the SY1-containing inclusion(s) in daughter cells is minimal (i.e., did not reach a required threshold) [39,40].

### 2.4. Disrupting Methylglyoxalase System and Aldose Reductase Activity Is Associated with Reduced SY1 Protein Content in S. cerevisiae

The observed difference in the percentage of fluorescence positive cells between wt and *glo1*∆ cells on the one hand, and *glo2*∆ and *gre3*∆ cells, on the other hand, prompted verification of the SY1 protein level via Western blot analysis. To this end, we subjected whole-cell extracts of wt, *glo1*∆, *glo2*∆, and *gre3*∆ *S. cerevisiae*, and wt, *glo1*∆, and *glo2*∆ *S. pombe* strains to Western blot analysis using an anti-SY1 antibody. Our analysis confirmed that *glo2*∆ and *gre3*∆ *S. cerevisiae* strains displayed a reduced SY1 protein content in comparison to wt and *glo1*∆ strains (Figure 5A), a result in line with the fluorescence microscopy data. This was surprising given the observations indicating negative cross-talk between glycation and ubiquitination of proteins, including αSyn [42,43].

Increased protein glycation has been shown to interfere with ubiquitin-proteasomal degradation as several ubiquitination sites are subjective to glycation as well. We detected protein bands most likely corresponding to SY1 aggregates above the 250 kDa protein marker after longer exposure times, in samples containing sufficient SY1 (indicated by black arrow).

In contrast to *S. cerevisiae*, we did not detect differences in SY1 protein level between the *S. pombe* strains (Figure 5B), a finding supported by the fluorescence microscopy analysis which showed no differences in the number of fluorescence-positive cells between the strains.

### 2.5. Potential Localisation of dsRed-SY1 Inclusions to IPOD in Methylglyoxalase System and Aldose Reductase-Deficient Strains Indicates the Involvement of Autophagy

Cytoplasmic quality control compartment (CytoQ), juxta/intranuclear quality control compartment (JUNQ/INQ), and insoluble protein deposit (IPOD) were identified in *S. cerevisiae* as intracellular compartments for the sequestration of misfolded cytosolic proteins [44,45]. While soluble misfolded proteins are sorted to CytoQ and JUNQ/INQ for refolding or proteasome-mediated chaperone-dependent degradation, insoluble terminally aggregated proteins are sequestered and degraded via autophagy in the vacuole. We observed predominantly large dsRed-SY1 aggregates in wt, *glo1*∆, *glo2*∆, and *gre3*∆ strains. Moreover, we detected an increased number of large aggregates in *glo2*∆ and *gre3*∆ cells, potentially the result of increased SY1 oxidation and glycation, respectively. Increased glycation of dsRed-SY1 interferes with ubiquitination activity and therefore with proteasomal degradation [42].

Therefore, we assessed the extent to which inclusions were perivacuolarly localised, indicative of SY1 inclusion localisation to IPOD. Preliminary quantification of the number of inclusions located either in the proximity of/or distant to the nucleus revealed that in the majority of *glo1*∆, *glo2*∆, and *gre3*∆ cells, dsRed-SY1 inclusions were distantly located. In contrast, inclusions in the majority of wt cells displayed perinuclear localisation (Figure 6A). Inclusions in the majority of cells of all strains were found in the proximity of the vacuole (Figure 6B). Combined, these findings indicate the involvement of both JUNQ and IPOD in degrading inclusions formed in wt cells, while SY1 inclusions in *glo1*∆, *glo2*∆, and *gre3*∆ cells seem to be degraded primarily via autophagy through sequestration in IPOD [45].

Thus, SY1-mediated toxicity in *glo2*∆ and *gre3*∆ cells is associated with an increased proportion of cells displaying multiple large aggregates.

### 2.6. Disrupting Methylglyoxalase System and Aldose Reductase Activity Is Associated with Enhanced Oxidative Stress Levels in S. cerevisiae

Next, we assessed the accumulation of ROS in the *SNCAIP*-expressing strains, as αSyn and SY1 aggregates are known to cause mitochondrial dysfunction, leading to the production of ROS and eventually cell death [14,46,47,48]. ROS accumulation could explain the observed growth-inhibiting effect in the *SNCAIP*-expressing *glo2*∆ and *gre3*∆ strains. Generation of ROS was detected using dihydroethidium (DHE), which can be used as a measure of the extent of superoxide and hydrogen peroxide formation in cells. Surprisingly, both the base level, i.e., upon expressing an empty plasmid, and the SY1-associated oxidative stress level detected in *glo2*∆ and *gre3*∆ cells were lower than the corresponding values detected in wt and *glo1*∆ cells. Contrary to wt and *glo1*∆ cells in which *SNCAIP* expression reduced or did not affect the percentage of positive cells, respectively, *glo2*∆ and *gre3*∆ strains showed a strong increase in the percentage of DHE-positive cells upon expression of *SNCAIP* when compared with the control (Figure 7). Thus, SY1-mediated toxicity in *glo2*∆ and *gre3*∆ is associated with an increased proportion of cells displaying multiple large aggregates and increased oxidative stress levels.

### 2.7. Analysis of Actin Cytoskeleton Organisation in S. cerevisiae Cells Producing dsRed-SY1

SY1 interacts with actin and affects actin cytoskeleton organisation in wt *S. cerevisiae* [14]. More specifically, Büttner et al. found that individual SY1 inclusions localise to actin cables and that they are usually flanked by one or more actin patches. This way, these authors suggested that the inclusions are able to incite actin cytoskeleton delocalisation to allow inclusion transport along actin cables. Polarisome-regulated retrograde transport along these actin polymers from daughter to mother cell serves as a defence mechanism by retaining aggregates in mother cells, resulting in a longer living progeny [15]. As such, retained aggregates are assembled in larger aggresomes and degraded by autophagy.

Due to the dual observation of a growth-inhibiting effect and enhanced SY1 inclusion formation in *glo2*∆ and *gre3*∆ cells, we wished to investigate actin morphology and the localisation of dsRed-SY1 inclusions to the actin cytoskeleton in a subsequent step.

F-actin forms three distinct filamentous structures during the cell cycle—patches, cables, and the actin ring. During interphase and towards the end of the mitotic cycle, patches appear mainly at the zone of cell growth and area of bud formation, respectively. Upon ceasing cell division, the actin cytoskeleton reorganises for the most part into so-called actin bodies which serve as reserves of F-actin that can be immediately mobilised upon re-entry into a proliferation cycle to form patches and cables [49,50]. Therefore, we analysed actin cytoskeleton organisation and SY1 localisation during the exponential (Figure 8A) and stationary (Figure 8B) growth phase.

During exponential growth, localisation of dsRed-SY1 inclusions adjacent to actin cables was visible in cells which displayed both normal (wt and *glo1*∆) and reduced (*glo2*∆ and *gre3*∆) growth upon *SNCAIP* expression. Moreover, co-localisation of dsRed-SY1 inclusions and actin cables was visible in actively dividing wt and *gre3*∆ cells, pointing towards retrograde transport of inclusions via actin cables to the mother cell. In addition, in both wt and *gre3*∆ cells, inclusions were flanked by actin patches, but not to the extent as described previously [14] (Figure 8A).

On the other hand, during the stationary phase, multiple potential actin bodies were observed in wt, *glo1*∆, *glo2*∆, and *gre3*∆ cells. These actin reserves were dispersed throughout the cell but primarily found near the cell membrane. In addition, cells of all, but especially of *glo2*∆ and *gre3*∆, strains contained a higher number of large inclusions during the stationary growth phase compared to cells in the exponential growth phase. In each case, at least one inclusion of each cell localised to one or more actin bodies (Figure 8B).

## 3. Discussion

PD is characterised by the presence of Lewy body inclusions consisting of aggregates of αSyn and other proteins such as SY1. Although increased levels of the glycating agent MGO have been detected in certain brain regions of PD patients, no direct causative link has been established yet between increased glycation and PD. On the other hand, the influence of glycation on αSyn aggregation ability has been established, including in an *S. cerevisiae* yeast cell model in which glycation was found to increase the accumulation of toxic αSyn oligomers in vitro and in vivo [35]. In the current study, we made an effort to gain a better understanding of the impact of a functionally disrupted methylglyoxalase system and aldose reductase activity on SY1 inclusion characteristics and cell growth. To this end, we employed *dsRed*-*SNCAIP*-expressing wt, *glo1*∆, *glo2*∆, and *gre3*∆ budding yeast, and fission yeast strains deleted for *glo1*^+^ and *glo2*^+^.

In this study, we only used *glo2*∆ to study the role of glyoxalase 2 activity, and not of its paralog *GLO4*. Similar to Glo2, Glo4 can hydrolyse S-D-lactoylglutathione. However, Glo4 is mitochondrial matrix-localised and expressed exclusively on the non-fermentable carbon source glycerol [28]. Because growth experiments were carried out on glucose-containing medium, glyoxalase 2 activity of Glo4 was not of relevance in our studies. In fact, the significance of mitochondrial glyoxalase 2 activity is questioned since no glyoxalase 1 activity is detected in yeast mitochondria [28]. In addition, even though S-D-lactoylglutathione was shown to enter the mitochondria of rat hepatocytes [51], there are to the knowledge of the authors of this article no studies reporting on the import of S-D-lactoylglutathione itself in *S. cerevisiae* mitochondria.

We observed an increase in the formation of large dsRed-SY1 inclusions upon disrupting Glo2 and Gre3 functionality. Moreover, this was associated with an increase in oxidative stress and a resulting growth defect. Oxidative stress is caused by ROS overproduction and points towards a disruption of mitochondrial function due to the accumulation of SY1 inclusions. In a feedback loop, ROS can then further enhance inclusion formation through oxidation of cysteine residues, which facilitates the formation of intra- and intermolecular disulphide bridges. dsRed-SY1 inclusions localised to actin cables in exponentially growing strains, including in actively dividing *gre3*∆ cells, and were primarily found perivacuolar, indicative of compartmentalisation in IPOD. These findings combined strongly indicate that dsRed-SY1 inclusions are subject to retrograde transport from daughter to mother cell where they are assembled into protective aggresomes and degraded via autophagy. However, we reason that the autophagic degradation capacity is rapidly overloaded in *glo2*∆ and *gre3*∆ cells due to enhanced SY1 inclusion formation and aggresome build-up. This could eventually affect mitochondrial integrity and lead to overproduction of ROS and an associated growth defect.

We conducted a comparative study to assess any potential repercussions of a disrupted methylglyoxalase system on the growth of the fission yeast *S. pombe* upon expression of *SNCAIP*. Similar to *S. cerevisiae*, the glyoxalase system, encoded by *glo1^+^* and *glo2^+^*, plays an important role in the detoxification of MGO [52]. Nevertheless, we did not observe repercussions on growth upon disrupting the function of Glo1, indicating that other pathways can metabolise the endogenous MGO levels. Indeed, *S. pombe* Hsp31 proteins (Hsp3101 and Hsp3102) displayed Glo3 activity and conferred MGO resistance in a *glo1*∆ strain [53]. Similar to *S. cerevisiae*, we did not observe growth defects in an SY1-expressing *glo1*∆ strain. This points towards a dominant role of Hsp31-mediated Glo3 activity in detoxifying MGO. As mentioned in the introduction, the glyoxalase 2 activity of the proposed *S. pombe* Glo2 homolog has yet to be shown [29].

The absence of an SY1-associated growth phenotype and similar levels and intracellular distribution of SY1 inclusions in *glo2*∆ cells in comparison to wt cells indicates a lack of glyoxalase 2 functionality of Glo2. Alternatively, disruption of Glo2 activity might be compensated for by the mitochondrial thioredoxin system, which was shown to act as a back-up system for glutathione-mediated antioxidant activity in fission yeast [54]. Interestingly, we consistently detected SY1 inclusions in the cell tips upon expression of *SNCAIP*.

This points towards the involvement of a transport mechanism actively regulating the intracellular localisation of these inclusions. In fission yeast, polarity factors, deposited at the plasma membrane by microtubule-mediated transport, recruit protein complexes (e.g., polarisome) involved in actin cable assembly [55]. These actin cables act as tracks for cargoes, including vesicles, to the cell tips [56]. It is therefore tempting to speculate that SY1 associates with inclusions that are subsequently transported to the cell tips through direct or indirect interaction with vesicles [6,14]. In contrast to *S. cerevisiae*, the observed inclusions in *S. pombe* did not show signs of fusion or retrograde transport via actin cables to facilitate asymmetric distribution into one of the two daughter cells. These findings were in line with the lack of observed SY1-associated growth defect.

## 4. Materials and Methods

### 4.1. Cell Culture, Strains, and Plasmids

*S. cerevisiae* strains were grown in 3 mL minimal medium, containing glucose (2%; Thermo Fisher Scientific, Waltham, MA, USA) and yeast nitrogen base without amino acids and ammonium sulphate (1.9 g/L; Formedium, Norfolk, UK) at 30 °C. Supplements were ammonium sulphate (5 g/L; VWR, Leuven, Belgium) and complete supplement mixture drop-out depleted for uracil (770 mg/L; Formedium). Cells were grown for 48 h after the time point of inoculation and subsequently subjected to the appropriate analysis. The wt and deletion strains *glo1*::*kanMx*, *glo2*::*kanMx*, and *gre3*::*kanMx* were BY4741 MATa, *his3∆1 leu2∆0 met15∆0 ura3∆0*. All strains were haploid. *SNCAIP*^WT^ and *dsRed*-*SNCAIP*^WT^ were expressed from a multicopy episomal plasmid (pYX212 backbone with *URA3* marker) using the triosephosphate isomerase (*TPI*) promoter. The cDNA encoding synphilin-1 was isolated from a hippocampal cDNA library via PCR amplification using the primers *CATGCCATGGAAGCCCCTGAATACC* and *CCGCTCGAGTTATGCTGCCTTATTCTTTCC* that included, respectively, a NcoI and XhoI restriction site for cloning into the pYX212 plasmid, which allows expression from the constitutive *TPI1* promoter [14].

*S. pombe* cell cultures were grown in Edinburgh Minimal Medium [57] (Formedium) supplemented with glutamic acid (20 mM; Sigma-Aldrich, St. Louis, MO, USA) (EMMG) as a nitrogen source, at a temperature of 25 °C. Unless specified otherwise, cells were grown and maintained at log phase in EMMG during, and for 48 h prior to, induction of gene expression. Crosses were carried out on EMM-N supplemented with thiamine hydrochloride (5 µM; Sigma-Aldrich) to repress nmt promoter expression, and amino acids were added to a concentration of 225 mg/L. The wt *S. pombe* strain used in the study was h^90^
*ura4.D18*. The deletion strains *glo1*::kanMX6 and *glo2*::kanMX6 were h^+^
*ade6-M210/M216 ura4-D18 leu1-32* (Bioneer *S. pombe* deletion mutant library [58]). All strains were haploid. *SNCAIP*^WT^ was cloned in the pINT plasmid backbone [59] (*ura4* marker) [60] and expressed using the thiamine repressible no message with thiamine (nmt1) promoter [60]. Expression plasmids were linearised prior to yeast transformation. Upon successful pINTL transformation and integration in the *leu1* locus, strains simultaneously acquire uracil prototrophy and leucine auxotrophy.

### 4.2. Growth Analyses and Spot Assays

*S. cerevisiae* and *S. pombe* strains were first grown as described in 2.1, after which the cells were inoculated at an optical density of 600 nm of 0.1 in either a 96-well (*S. cerevisiae*) or 24-well (*S. pombe*) plate (CELLSTAR, Greiner Bio-One, Kremsmünster, Austria) for subsequent growth in a Multiskan GO microplate spectrophotometer (Thermo Scientific). Plates were shaken at 400 rpm and optical density was measured at 600 nm every 30 min. For spot assays, serial dilutions of *S. cerevisiae* and *S. pombe* precultures (OD values of 0.5–0.05–0.005–0.0005) were spotted onto the appropriate solid minimal medium (see 2.1). Two to four days after spotting, colonies appeared and plate scans were made. Growth profiles are based on three clones per strain.

### 4.3. Oxidative Stress Measurements

*S. cerevisiae* strains were grown in a 96-well plate as described in 2.1. Accumulation of ROS was analysed based on the superoxide-driven conversion of non-fluorescent dihydroethidium (DHE) to fluorescent ethidium. The DHE assay was carried out as described in the guidelines (ThermoFisher) and quantified using flow cytometry (Guava easyCyte 8HT, Merck Millipore, Burlington, MA, USA). A two-way ANOVA test (*p*-value of 0.05) was carried out using GraphPad Prism 8.3.0 (GraphPad Software, San Diego, CA, USA).

### 4.4. Fluorescence Microscopic Analysis

SY1 inclusion formation and effects on cytoskeletal architecture in *S. cerevisiae* were analysed by standard fluorescence microscopy using a Leica DM4000B (Leica, Wetzlar, Germany). To assess differences between strains in inclusion characteristics, a comparative analysis based on at least 800 cells originating from three clones per strain was performed. Only fluorescence positive cells were considered for the comparative analysis of the number and size of inclusions between strains. Statistical analysis (one or two-way ANOVA test, *p*-value of 0.05) was carried out using GraphPad Prism 8.3.0.

Cell nuclei were stained using DAPI (Thermo Fisher). To this end, cells were fixed in 37% formaldehyde for 1 h at room temperature, after which the cells were washed twice in PBS. Nuclei were subsequently stained in 1:3 DAPI/PBS for 1 h in the dark. Cell vacuoles were stained using CellTracker^TM^ Blue CMAC dye (Thermo Fisher) according to the manufacturer’s guidelines.

Finally, cells were washed trice in PBS and subjected to microscopic analysis. Quantifications are based on the localisation of 100 inclusions in around 50 cells of one clone per strain, grown until the mid-late exponential phase. Due to limitations in the number of independent cell countings, no statistics were performed for the inclusion localisation experiment. The actin staining was performed using Alexa Fluor^TM^ 488 phalloidin (Thermo Fisher) and conducted as recommended in the kit’s guidelines.

### 4.5. Immunological Techniques

#### 4.5.1. Western Blot Analysis

*S. cerevisiae* whole-cell extracts were prepared according to Vandebroek et al [61], while *S. pombe* whole-cell extracts were prepared according to Hagan et al. [62]. Proteins were separated by SDS−PAGE on 10% Tris-glycine gels and transferred to PVDF membranes (0.45 µm; Merck Millipore). SY1 and tubulin were detected using rabbit anti-SY1 antibody (1 mg/mL, Sigma Aldrich) and rat anti-tubulin antibody (1 mg/mL, Abcam, Cambridge, UK), respectively. Secondary antibodies used were mouse anti-rabbit IgG-HRP and mouse anti-rat IgG-HRP (0.4 mg/mL and 1 mg/mL, respectively, Santa Cruz Biotechnology, Dallas, TX, USA). Primary and secondary antibody solutions were diluted 1:1000 and 1:10,000 in 5% milk/TBS-T, respectively.

A one-way ANOVA test (*p*-value of 0.05) was conducted to assess differences in anti-SY1 antibody immunoreactivity between strains. Standard deviations are based on densitometric data, as determined by Fiji Software (v1.52i), of three clones per strain.

#### 4.5.2. Immunofluorescence Microscopy

Immunofluorescence microscopy was conducted using a Leica DMi8 (Leica, Wetzlar, Germany) inverted microscope. Cell fixation, permeabilisation, and antibody application steps were performed according to the protocol described in [63], omitting glutaraldehyde addition and quenching steps. Primary antibody anti-SY1 (1 mg/mL, Abcam) and goat anti-rabbit secondary antibody coupled to the red Alexa fluor 594 dye (2 mg/mL, Thermo Fisher) were used in 1:50 and 1:100 dilutions, respectively.

## 5. Conclusions

In this study, we investigated the impact of a functionally disrupted glyoxalase system and aldose reductase activity on SY1 inclusion characteristics and cell growth in the yeasts *S. cerevisiae* and *S. pombe*. We found that lack of Glo2 and Gre3 activity in *S. cerevisiae* affected both SY1 inclusion number and growth ability. Increased SY1 inclusion formation correlated with enhanced oxidative stress levels and an inhibitory effect on growth, most likely caused by deregulated autophagic degradation due to excess aggresome build-up. These findings illustrate the severe repercussions of enhanced oxidation and glycation on SY1 inclusion formation.

Future endeavours could focus on co-expressing SY1 and αSyn to shed more light on their interplay in conditions characterised by increased protein glycation and oxidation. Moreover, the use of double mutants such as *glo1*∆ *gre3*∆ could help unravel the connection between both MGO detoxification pathways.

Finally, while we clearly detected inclusion formation in our fission yeast model, the level of incited stress upon *SNCAIP* expression did not seem to reach a sufficiently high level to impair the growth of any of the tested strains. In addition, our data do not provide any indications that the fission yeast homolog of *GLO2*, *glo2*^+^, carries out glyoxalase 2 activity. Combined, our findings illustrate the relevance of yeasts, especially *S. cerevisiae*, as complementary model organisms to unravel mechanisms contributing to SY1 protein pathology in the context of neurodegenerative diseases.

## Figures and Tables

**Figure 1 ijms-22-01677-f001:**
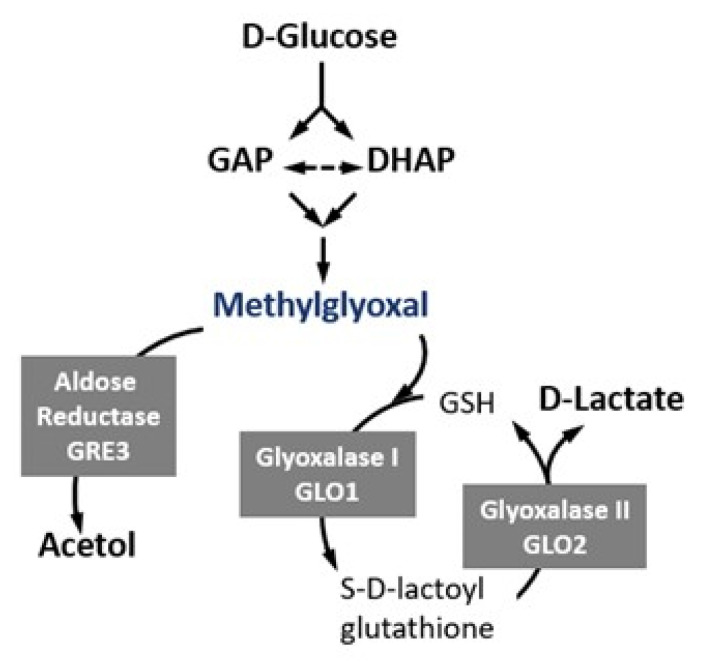
Overview of methylglyoxal (MGO) metabolism. MGO, the most reactive glycating agent is primarily formed from the spontaneous decomposition of the phosphate group of glyceraldehyde 3-phosphate (GAP) and dihydroxiacetone phosphate (DHAP). It may also arise from the oxidation of aminoacetone in the catabolism of l-threonine from the oxidation of ketone bodies and the oxidation of acetone. MGO is catabolised via the glyoxalase system (Glo1 and Glo2), resulting in the formation of D-lactate and glutathione, and by aldose reductase Gre3, resulting in acetol formation.

**Figure 2 ijms-22-01677-f002:**
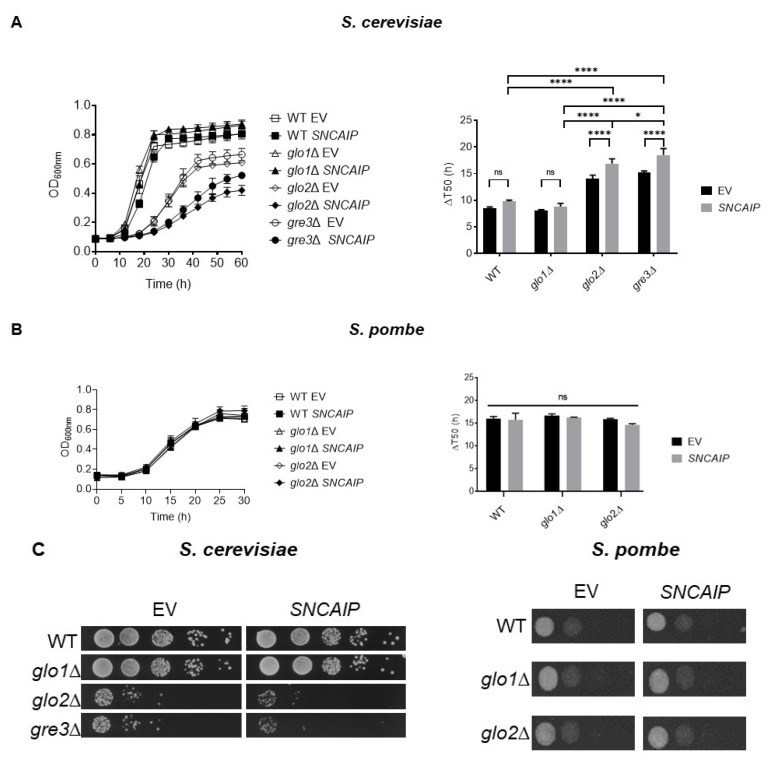
Analysis of the repercussions of a disrupted methylglyoxalase system and aldose reductase activity on the growth of *SNCAIP*-expressing *S. cerevisiae* (**A** and **C** left) and *S. pombe* (**B** and **C** right). ∆T_50_ indicates the number of hours needed for a strain to reach the OD value corresponding to half the OD_max_ value. Data represent the combined results of three clones per strain. Error bars represent the standard deviations based on the obtained data of these clones. Significance is determined using a one-way ANOVA test (*p*-value of 0.05, significance indicated by the asterisk; *n* = 3, * = *p* < 0.05, **** = *p* < 0.0001). Non-indicated differences between strains are not significant. Growth of *S. cerevisiae* and *S. pombe* was measured using 96-well and 24-well plates, respectively. EV = empty vector; ns = not significant.

**Figure 3 ijms-22-01677-f003:**
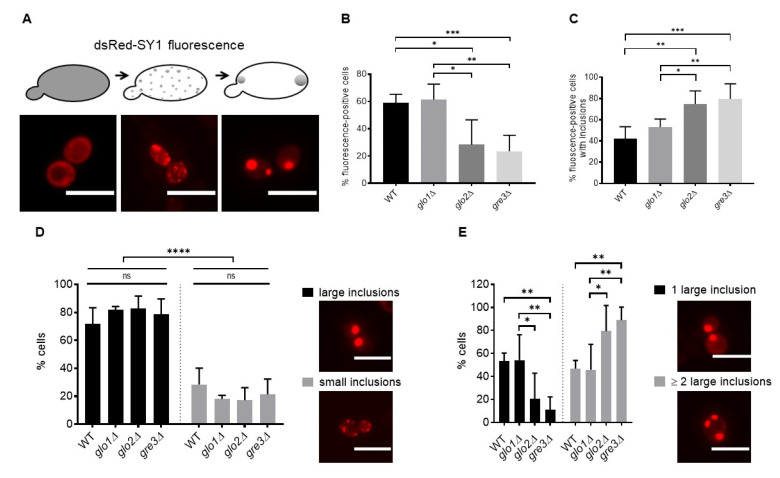
Analysis of the repercussions of a disrupted methylglyoxalase system and aldose reductase activity on SY1 inclusion characteristics in *S. cerevisiae*. dsRed-SY1 forms consecutively small oligomers and larger membrane-localised aggregates in wt *S. cerevisiae* [14] (**A**). The percentage of wt, *glo1*∆, *glo2*∆, and *gre3*∆ cells displaying dsRed-SY1 fluorescence (**B**), dsRed-SY1 fluorescence and inclusions (**C**), small or large dsRed-SY1 inclusions (**D**), and multiple large dsRed-SY1 inclusions (**E**) was calculated. Data represent the combined results of at least 800 counted cells of three clones per strain. Error bars represent the standard deviations based on these counts. Significance is based on the total number of cells and determined using a one or two-way ANOVA test (*p*-value of 0.05, significance indicated by the asterisk; *n* = 3, * = *p* < 0.05, ** = *p* < 0.01, *** = *p* < 0.001, **** = *p* < 0.0001). Non-indicated differences between strains are not significant. The scale bars represent 10 µm. SY1 = Synphilin-1; ns = not significant.

**Figure 4 ijms-22-01677-f004:**
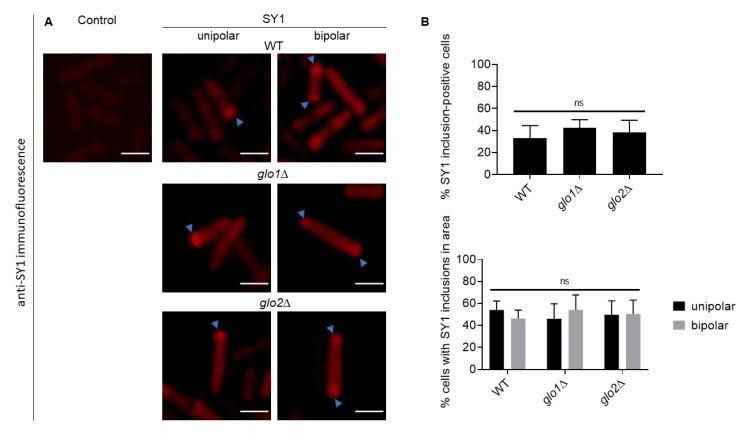
Analysis of the impact of a disrupted methylglyoxalase system on SY1 inclusion characteristics in wt, *glo1*∆, and *glo2*∆ *S. pombe* strains. Native untagged SY1 was visualised via immunofluorescence microscopy using an anti-SY1 antibody (**A**), and the percentage of SY1-positive cells and the percentage of cells with unipolar or bipolar-localised inclusions were quantified (**B**). Significance is determined using a one or two-way ANOVA analysis (*p*-value of 0.05). The scale bars represent 10 µm. SY1 = Synphilin-1; ns = not significant.

**Figure 5 ijms-22-01677-f005:**
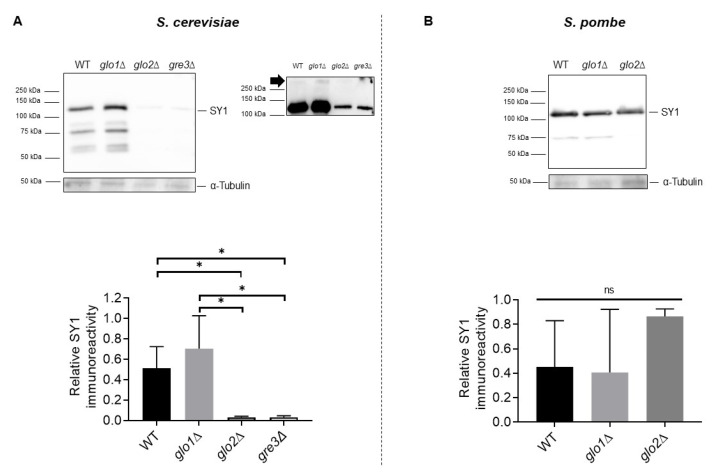
Analysis of SY1 protein level in wt, *glo1*∆, *glo2*∆, and *gre3*∆ *S. cerevisiae* (**A**) and *S. pombe* (**B**) strains. Protein bands corresponding to SY1 were clearly visible after 30″ exposure time (panel A top left), while bands, most likely corresponding to aggregated species, appeared above the 250 kDa protein marker around 2′30″ exposure time (panel A top right, indicated with black arrow). SY1 protein level quantifications are based on densitometric data of three clones per strain and are normalised using the total protein content as determined by anti-α-Tubulin antibody ‘TAT-1′. Error bars represent the standard deviations based on densitometric data of these clones. Significance is determined using a one-way ANOVA test (*p*-value of 0.05, significance indicated by the asterisk; *n* = 3, * = *p* < 0.05). Non-indicated differences between strains are not significant. SY1 = Synphilin-1.

**Figure 6 ijms-22-01677-f006:**
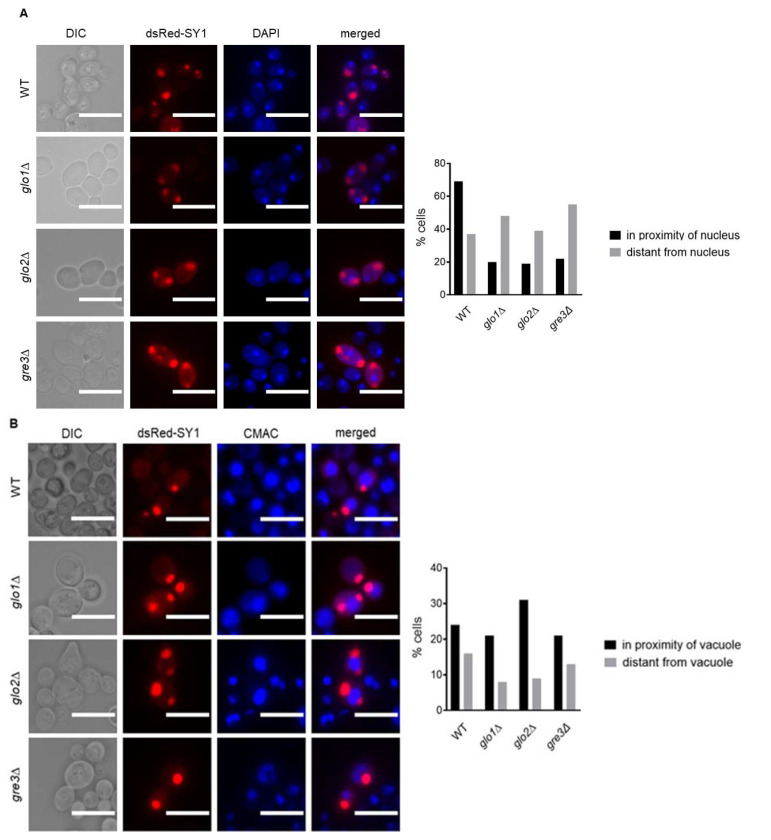
Nuclear (**A**) and vacuolar (**B**) staining of *dsRed*-*SNCAIP*-expressing *S. cerevisiae* wt, *glo1*∆, *glo2*∆, and *gre3*∆ strains. A preliminary quantification of dsRed-SY1 inclusion localisation was performed based on 100 inclusions in less than 50 cells of one clone per strain. The scale bars represent 10 µm. SY1 = Synphilin-1.

**Figure 7 ijms-22-01677-f007:**
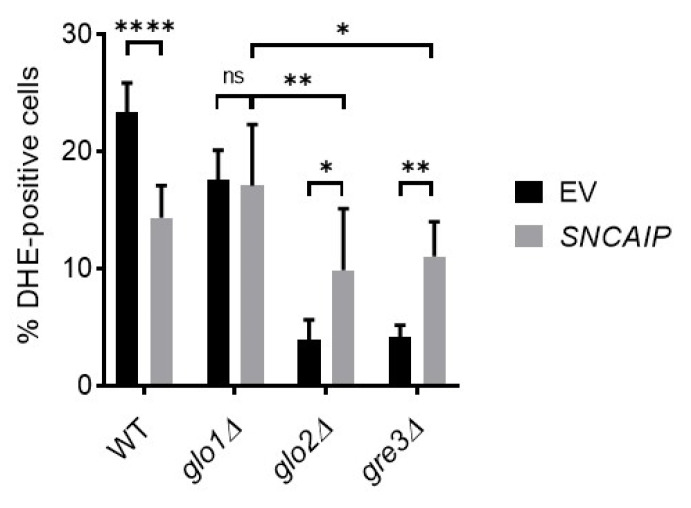
Impact of a disrupted methylglyoxalase system and aldose reductase activity on SY1-associated oxidative stress in *S. cerevisiae*. Calculated percentages of dihydroethidium (DHE)-positive cells indicate the extent of induced oxidative stress in each strain. Data represent the combined results of cells of five clones per strain. Error bars represent the standard deviations based on DHE-positive cell counts of these clones. Significance is based on the total number of cells counted and determined using two-way ANOVA analysis (*p*-value of 0.05, significance indicated by the asterisk; *n* = 5, * = *p* < 0.05, ** = *p* < 0.01, **** = *p* < 0.0001). Non-indicated differences between strains are not significant. DHE = dihydroethidium; EV = empty vector; ns = not significant.

**Figure 8 ijms-22-01677-f008:**
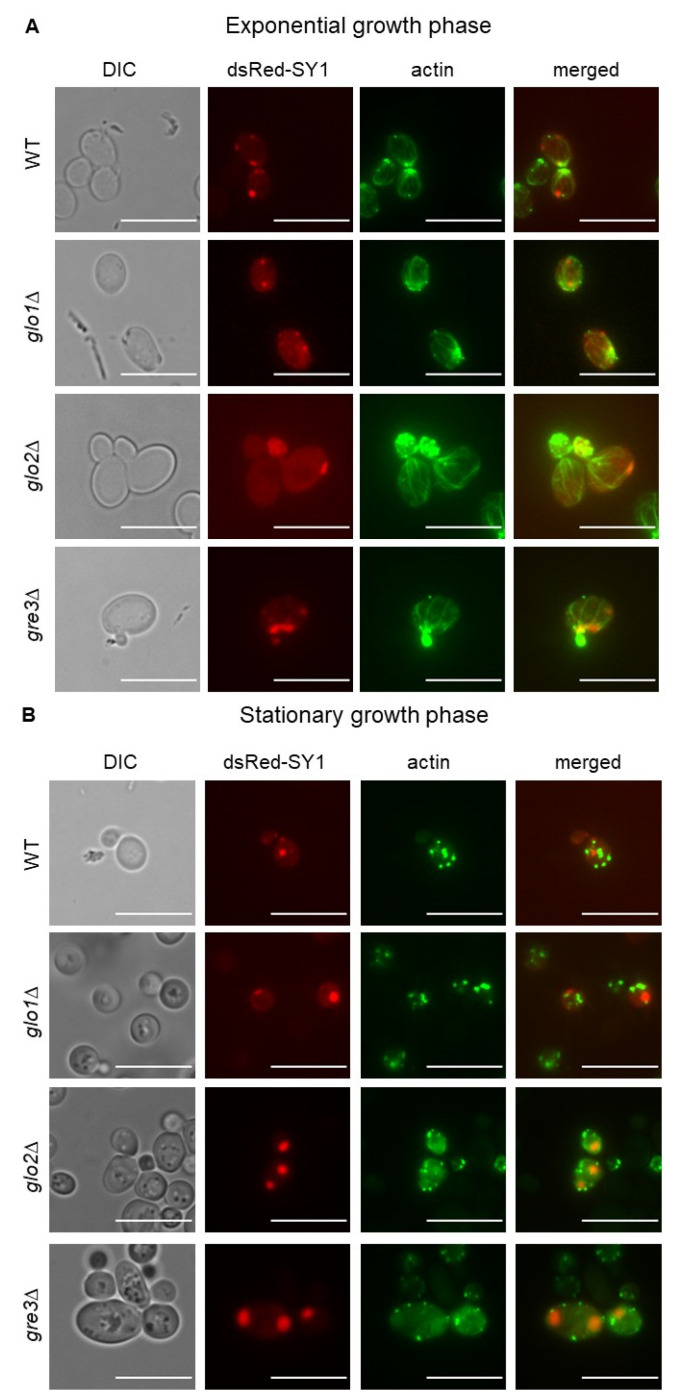
Microscopic analysis of actin cytoskeleton organisation and SY1 inclusion localisation in *dsRed-SNCAIP*-expressing *S. cerevisiae* strains during exponential (**A**) and stationary (**B**) growth phase. The scale bars represent 10 µm. SY1 = Synphilin-1.

**Table 1 ijms-22-01677-t001:** Overview of the functions of Glo1, Glo2, and Gre3 in *Saccharomyces*
*cerevisiae* and the corresponding orthologs in *Schizosaccharomyces*
*pombe* and humans.

		Glo1	Glo2	Gre3
***S. cerevisiae***	gene	*GLO1* (Lactoylglutathione lyase)	*GLO2* and *GLO4* (Hydroxyacylglutathione hydrolase)	*GRE3* (NADPH-dependent aldose reductase)
	function	catalyzes the formation of S-D-lactoylglutathione	catalyzes the hydrolysis of S-D-lactoyl-glutathione to form glutathione and D-lactic acid	reduces methylglyoxal to acetol and (R)-lactaldehyde
***S. pombe* Ortholog**	gene	*glo1^+^* (Lactoylglutathione lyase)	*glo2^+^* (Hydroxyacylglutathione hydrolase)	non existing
	function	the same as *S. cerevisiae* Glo1	function not experimentally validated yet	no *S. pombe* aldose reductase has reported MGO-detoxifying activity
**Human Ortholog**	gene	*GLO1* (Lactoylglutathione lyase)	*GLO2* (Hydroxyacylglutathione hydrolase)	*AKR1B10* (aldo-keto reductase family 1 member B10)
	function	the same as *S. cerevisiae* Glo1	the same as *S. cerevisiae* Glo2	catalyzes the NADPH-dependent reduction of a wide variety of carbonyl-containing compounds

## Data Availability

The data presented in this study are available on request from the corresponding author.

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
