# Peer review of "Yeasts as Complementary Model Systems for the Study of the Pathological Repercussions of Enhanced Synphilin-1 Glycation and Oxidation"

_ijms, 2021, doi:10.3390/ijms22041677_

Round 1
Reviewer 1 Report
In this ms. Seynnave et al. investigate the outcome from heterologous expression of human Synphilin-1 (SY1) protein, an interaction partner of α-Synuclein, in the model organisms budding and fission yeasts. In particular, the authors searched for deleterious effects in cells lacking the glyoxalase and aldose reductase functions. Among others, Seynnave et al. find that overexpressing SY1 is toxic in budding yeast cells lacking Glo2 and Gre3, presumably by an increase in SY1 inclusions. This observation also correlates with an enhanced oxidative stress in such cells expressing SY1. Overall, the presented findings are interesting and merit publication, however the quality of the presentation must be considerably improved. Please find my comments and suggestions below.
Abstract:
Please clarify that the human SY1 protein is used in this study and that DsRed is a fluorescent protein.
Figure 1:
Adding a small table describing the gene names and functions of GLO1, GLO2, GRE3 in Sc, Sp and humans would be useful.
Figure 2:
_ There is no correlation between text, legend and figure as panel 2C is not acknowledged anywhere.
_ The spotting assay showing toxicity in glo2∆ and gre3∆ cells could be more convincing if the authors include another assay using more starting cells and 5-fold dilutions.
_ Including a glo4∆ as negative control would be useful.
_ Please state in the legend that 96-well plates were used instead of conventional 10mm cuvettes.
Figure 3:
_ There is no description of the criteria used to define small and large inclusions. Is it just visual observations?
_ Panels D & E: Is the counting restricted to fluorescent-positive cells?
Figure 5:
_ It is unclear why the WB is so different than microscopy, despite the fact that the authors state a correlation (lines 284-286). Basically, I notice a >10-fold reduction in protein levels while the amount of fluorescent cells is about half (Fig. 3B) This must be investigated more in detail. For instances, what is the outcome from measuring total DsRed fluorescence with a microplate reader?
_ RT-PCR confirming that SY1 mRNA levels are similar would be useful.
Language:
_ Please use active voice and avoid present tense to describe observations (e.g. lines 367-368).
_ Please carefully scrutinize the text as awkward sentences can be found (e.g. Line 70: “The identified… been experimentally identified”). Similarly, it is unclear which authors are referred in lines 46, 462 and 573.
_ Line 51: “Sir-2” must read “Sir2”.
Author Response
Abstract remark: suggestion incorporated
Fig.1 remark: suggestion incorporated
Fig.2 remarks 1 and 4: suggestions incorporated
Fig.2 remark 2: I agree. Nevertheless, the growth difference between wt/glo1del and glo2del/gre3del, especially in combination with the growth curve analysis, is clear.
Fig.2 remark 3: I agree. On the other hand, not having this control included does not change the validity of the results currently included; Glo4 is expressed exclusively on the non-fermentable carbon source glycerol.
Fig.3 remark 1: yes, there were no quantitative parameters included to differentiate between small and large inclusions. This was based on visible observation.
Fig.3 remark 2: yes
Fig.5: In this paper, we wanted to characterize the impact of increased glycation/oxidation on inclusions characteristics and document the repercussions on growth. Although I completely agree that the suggested experiments would create more insight, they do not change the key message we want to communicate. Therefore, I think that the suggested experiments are more suited for a follow-up article.
Language remarks: suggestions incorporated
Reviewer 2 Report
Seynnaeve et al., investigated the consequences of a disrupted glyoxalase system (deltaGlo1 or 2) or aldose reductase (deltaGre3) function on Synphilin-1 (a component of Lewy bodies, a hallmark of PD) inclusion formation, protein expression level, and cell growth using 2 yeast systems: Sc and Sp. ROS level and cell morphology were evaluated using Sc. While the Sc data has merit, the Sp data is not convincing.
Major
- What is the accession number of SY1? Is it codon optimized for yeast expression?
-Homology of Sc Glo1, 2 and Gre3; Sp Glo1 and 2 to human; Sc Glo2 to Sp; may be a figure?
-Add Sc and Sp symbols on graphs in fig2 and 5 could be helpful
-Fig 2c needs caption and citation in text (line 134, 139?). right panels of fig2A and B was not described in text
-The polar-localized inclusions signal in Sp is not that convincing (Fig 4)
-Fig5, is it more direct to do a Western with anti-dsRed antibody with Fig3?
-Fig6, what stage of cell cycle are those cells in? error bars? May low expression of dsRed-SY1 in dGLO1 and 2 strain contribute to an over-representation of the %?
-Line 321 “preliminary quantification”? Is there more analysis to be done?
-Fig8, how about a CoIP experiment with dsRED and F-actin, so you can quantify?
-Why is Sc SY1 expression plasmid multicopy episomal while Sp is integrated? May that contribute to the Sp results?
-How fast do you shake the 96wp or 24wp?
-What to expect in double or triple deletions?
Minor
-line 283: Western
Author Response
remark 1: the synphilin-1 cDNA was isolated from a hippocampal cDNA library via PCR amplification using the primers CATGCCATGGAAGCCCCTGAATACC and CCGCTCGAGTTATGCTGCCTTATTCTTTCC that included, respectively, a NcoI and XhoI restriction site for cloning into the pYX212 plasmid, which allows expression from the constitutive TPI1 promoter.
remark 2: suggestion incorporated
remark 3: suggestion incorporated
remark 4: suggestion incorporated: lines 144-146 + 166
remark 5 (Fig.4): there is a clear difference in fluorescence intensity visible when comparing the poles of the cell (indicated with the blue arrows) with the rest of the cytoplasm. I agree that our fluorescence microscopic result indicates, rather than proves, polar localization of SY1. Nevertheless, our argument is strengthened by literature describing SY1 interaction with vesicles and actin cable-mediated transport of said vesicles to the cell tips in fission yeast (Motegi et al., 2001 + Takahashi et al., 2006).
remark 6 (Fig.5): for the western blot analysis, we chose to work with native untagged SY1 constructs to avoid any potential interference of the dsRed tag with antibody recognition.
remark 7 (Fig.6): these cells were in mid-late exponential growth phase. Error bars were not included due to the limited number of independent countings we performed. Countings were restricted to the number of fluorescence-positive cells, so lower expression of dsRed-SY1 in glo2del and gre3del strains does not affect the result presented in Fig.6.
remark 8: we chose to use the word 'preliminary' since we only performed a limited number of independent countings (see previous comment). Our counting series did not allow us to perform statistics (and thereby strengthen our dsRed-SY1 localization claims).
remark 9 (Fig.8): that would indeed provide additional information and strenghten our claims regarding actin-mediated transport. Given the published data on actin cable-mediated transport of dsRed-SY1 inclusions and subsequent autophagic degradation (Büttner et al., 2010), it is, however, reasonable to draw conclusions from the observations currently presented in the article.
remark 10: there is no specific reason why we chose to work with multicopy episomal vs integrative plasmids. For Sc, we decided to continue to work with the same constructs used for the studies described in (Büttner et al., 2010). In the case of Sp, we were familiar with the pINT backbone from previous research.
We can not rule out that results are influenced by this different expression approach, but we believe that the chance is highly unlikely. Furthermore, the majority of our conclusions are based on inter-strain differences for each of the two yeast species, rather than on inter-yeast species differences.
remark 11: plates were shaken at 400rpm.
remark 12: In Sc, we expect that a double deletion of both GLO2 and GRE3, but not a double deletion of GLO1 and GLO2 or GLO1 and GRE3, would result in an exacerbated growth defect. In Sp, we do not expect a growth defect upon combined deletion of glo1+ and glo2+.
Round 2
Reviewer 1 Report
The manuscript has been improved and it is now suitable for publication.
Author Response
-spelling and conclusion section was improved
-M&M section was updated with more details on the methodology.
Reviewer 2 Report
I think the authors remarks to my comments is good enough and should be included in the materials and methods section of the article. These detailed information my be important for other researchers to repeat the experiments as they wish.
Author Response
-M&M section was improved by incorporating the additional details requested during review round 1.
-Spelling was improved.
-Conclusion section was improved.